# BabyBoom: 3-Dimensional Structure-Based Ligand and Protein Interaction Prediction by Molecular Docking

**DOI:** 10.3390/biom12111633

**Published:** 2022-11-03

**Authors:** Sameera Sastry Panchangam

**Affiliations:** Bioclues.org, Hyderabad 500072, India; sameera@bioclues.org; Tel.: +1-42-5772-4298

**Keywords:** Baby Boom (BBM), protein-protein interaction, docking, peptide mimetics, ligand-protein interactions

## Abstract

Baby Boom (BBM) is a key transcription factor that triggers embryogenesis, enhances transformation and regeneration efficiencies, and regulates developmental pathways in plants. Triggering or activating BBM in non-model crops could overcome the bottlenecks in plant breeding. Understanding BBM’s structure is critical for functional characterization and determination of interacting partners and/or ligands. The current in silico study aimed to study BBM’s sequence and conservation across all plant proteomes, predict protein-protein and protein-ligand interactions, and perform molecular docking and molecular dynamics (MD) simulation to specifically determine the binding site amino acid residues. In addition, peptide sequences that interact with BBM have also been predicted, which provide avenues for altered functional interactions and the design of peptide mimetics that can be experimentally validated for their role in tissue culture or transformation media. This novel data could pave the way for the exploitation of BBM’s potential as the master regulator of specialized plant processes such as apomixes, haploid embryogenesis, and CRISPR/Cas9 transgenic development.

## 1. Introduction

Transformation, embryogenesis, and plant regeneration are the key developmental stages in any plant improvement or breeding program. The advent of gene editing (GE) tools such as zinc finger nucleases (ZFNs), transcription-activator-like effector nucleases (TALENs), and clustered regulatory interspaced short palindromic repeats associated with protein 9 (CRISPR/Cas9) has enabled precise gene editing both in vitro and in planta leading to significant developments in crop improvement programs [1,2]. Although these tools aided crop improvement efforts regarding yield, quality, biotic and abiotic stress tolerance, and herbicide resistance, as reviewed by Gao [3] and Miladinovic [4], the success has been limited to a few crops such as rice, wheat, maize, barley, rapeseed, tomato, and watermelon. Many important food and cash crops such as legumes [5], nut and tree species, oilseeds (cotton and sunflower) [6], and most ornamental plants [7] are recalcitrant to GE, where plant regeneration is a major bottleneck for efficient deployment of GE techniques for breeding programs [3,8]. Genotype dependency, polyploidy, and complicated or lengthy transformation protocols further hinder the progress of GE in major crops. To achieve the full potential of GE, methods that enable faster, cheaper, at-scale production of edited plants are needed [9]. In order to overcome these bottlenecks, research is rapidly exploring methods such as co-delivery of developmental (DR) or morphogenetic regulators (MR) with CRISPR Reagents [10,11], viral vectors, and mobile RNAs for systemic delivery [12], biolistic delivery [13], and nanocarrier-mediated delivery of CRISPR/Cas reagents [14].

DRs have been extensively researched to enhance transformation efficiency and induce meristematic or embryogenic division [15,16,17,18]. Of the many DRs that help realize plasticity and totipotency, SOMATIC EMBRYOGENESIS LIKE RECEPTOR KINASE (SERK), LEAFY COTYLEDON (LEC), AGAMOUS-LIKE 15 (AGL15), Wuschel (WUS), and BABY BOOM (BBM) are known to be crucial [19]. More specifically, regeneration frequency and transformation efficiency were reported to be enhanced by Baby Boom (BBM) and Wuschel2 (WUS) in both dicot and monocot plants [10,20,21]. Taking advantage of this phenomenon, Agrobacterium cultures co-delivering BBM and WUS, and gene-editing cassettes were directly delivered into soil-grown plants and somatic tissues to enable tissue culture-free GE [15,21]. Boutilier et al. demonstrated that ectopic expression of BBM could dramatically promote organogenesis leading to the formation of somatic embryos in maize and when transgenes were co-delivered with DRs, transgenic embryos formed in 2 weeks and plantlets in 2 to 4 weeks [22]. Similar studies were carried out in sorghum [23], Arabidopsis, and Nicotiana [17,24].

BBM, a member of the AP2/ERF family and a key regulator of plant cell totipotency, was identified during the in vitro microspore embryogenesis of Brassica napus [22,25,26]. The spontaneous induction of embryogenesis and hormone induction of BBM has tremendous potential in specialized plant processes such as haploid and apomixes induction, which are complex, often requiring substantial personnel, equipment, and expertise, and are highly-genotype dependent [2,27]. Extensive genomic and transcriptomic studies have been conducted on BBM in a variety of crops to determine and establish its function and the genes it regulates. Despite these advances and potential applications limited information is available on BBM’s role and mechanism of action during embryogenesis [28]. Deciphering the structure of BBM protein and predicting its interactions is critical for functional characterization and determination of interacting partners and/or ligands. In the absence of an experimentally determined 3D structure, in silico modelling has progressed to yield significant structures in recent times. The impact of the prediction is two-fold, the first being new insights into the biochemical pathway of BBM, while the second is the identification of chemical analogues or protein mimetics that possess the same function. In addition, predicting accurate BBM-ligand binding or affinities could enhance fundamental knowledge of BBM’s mechanism and provide opportunities to improve tissue culture and transformation protocols by including the most promising ligands/biomolecules in the media. The current work presents in silico characterization of BBM with a focus on predicting interacting partners based on its 3-dimensional structure. This study aims to provide BBM’s structural and ligand interaction data, which would be the premise for designing experimental methods, which include Coimmunoprecipitation assays for establishing physical protein interactions, small-scale production of AtBBM in tobacco cell lines, isolation, purification, and X-ray crystallography for determining the 3-D structure of BBM. An out-of-the-box application is to procure the peptides designed in this study and use them in culture media to study the effect on embryogenesis and/or transformation efficiency. Such information will be transformative in plant breeding, allowing rapid production of gene-edited and haploid plants at lower costs and scale.

## 2. Materials and Methods

### 2.1. Orthology, Sub-Cellular Localization, and Domain Conservation

Complete plant proteome sequences were downloaded from Phytozome (https://phytozome-next.jgi.doe.gov/ (accessed on 3 February 2022)), which consists of 278 genomes [29]. A local protein sequence database was created using Blast+, and Arabidopsis thaliana BBM (AtBBM, Q6PQQ4.2) was used as a query to run Blastp 2.9.0+ locally with an e value of 1 × 10^−5^ [30]. The resultant orthologous sequences were submitted to WoLFPSORT, which makes predictions based on known sorting signal motifs and correlative sequence features such as amino acid content [31]. Further, JalView 2.11.1.7 (https://www.jalview.org/ (accessed on 3 February 2022)) [32] was used to predict domain conservation of BBMs and BBM-like proteins across the above plant proteins using MAFFT, and ClustalW alignment was exported to iTOL (https://itol.embl.de/ (accessed on 6 February 2022)) to generate a rooted circular Newick tree [33].

### 2.2. Protein-Protein and Protein-Ligand Interactions

Protein-protein interactions (PPI) are key to deciphering protein functions and providing insights into biochemical and/or metabolic processes. For this study, two approaches were employed to predict BBM PPIs: first via sequence-based prediction tools and the second based on the 3-dimensional structure of BBM. AtBBM was searched in the STRING v11.5 Database (https://string-db.org/) (accessed on 23 January 2022), and the interacting proteins with scores ≥ 0.90 were selected as primary interacting proteins [34]. In addition to STRING, BioGRID (https://thebiogrid.org/) (accessed on 23 January 2022) was also used to predict interactions [35]. For the structure-based predictions, 3-D structures of AtBBM and the interacting proteins were then searched individually in the UniProt database, and their structures were downloaded from the Alpha Fold server (https://alphafold.ebi.ac.uk/) (accessed on 15 March 2022) [36]. The continuous fragments of the structures predicted by AlphaFold with a confidence level > 90 were considered for further study. In this study, it was found that BBM had two separate fragments with a confidence score > 90. These two fragments of BBM protein were separately prepared. Similar structure selection and preparation methodologies were applied to BBM primary interacting proteins.

Predicting ligands that have an affinity for BBM can further elucidate its functional mechanisms. Protein-ligand binding sites and potential ligands were predicted using COACH (https://zhanggroup.org/COACH/) (accessed on 27 January 2022), a meta-server online prediction tool, which recognizes ligand-binding templates from the BioLiP protein function database by binding-specific substructure and sequence profile comparisons [37,38]. The tool combines results from other methods (including COFACTOR, FINDSITE, and ConCavity) available on the parent server https://zhanggroup.org/ (accessed on 27 January 2022). C-Sore is the confidence score of the prediction, which ranges between (0–1), where a higher score indicates a more reliable prediction.

### 2.3. 3-D Structure Preparation

AtBBM Agamous-like MADS-box protein (AGLI5), Nuclear transcription factor Y subunit B-9 (LEC1), Protein WUSCHEL (WUS), and B3 domain-containing transcription factor (LEC2) were the interacting proteins with an interaction score ≥ 0.9. Table 1 shows the resultant primary interacting proteins, corresponding UniProt IDs, and the interaction score. These four interacting proteins (AGLI5, LEC1, WUS, and LEC2) were then searched individually in the UniProt database, and their respective 3D structures were sourced from AlphaFold.

Fragment 1 of BBM contained residues numbered 208–217, 221–229, 243–268, 271–285, 295–296, and fragment 2 contained residues 316–317, 324–328, 339–358. Similar structure selection criteria were applied to BBM’s interacting proteins. Herein, AGLI5 also contained two fragments with a confidence score > 90, and thus they were separately prepared. The residues in AGLI5 fragment 1 were (4–69), and fragment 2 were (89–107, 113–165). However, LEC1, WUS, and LEC2 had a single fragment with a prediction confidence score > 90. All these fragments were extracted from the complete 3D structure of the protein. Figure 1 shows the structures of these fragments used in docking as receptors and ligands.

### 2.4. Docking and Clustering

BBM was docked with its interacting partners using a standalone application of Hex (http://hex.loria.fr/) (accessed on 18 May 2022). A standard docking protocol was followed where the correlation type was selected as “Shape + Electro + DARS”. The grid dimension that refers to sampling grid size was set at 0.6, solutions as 2000, receptor range (angle of scanning at protein surface)180°, ligand range (angle of scanning at ligand surface) 180°, twist range (intermolecular twist angle) 360°, distance range (limit of intermolecular separation from an initial distance) at 40, translation step as 0.8, and score threshold at 0.0. It was observed that BBM had two continuous stretches of amino acids responsible for DNA binding, and both fragments were docked sequentially with their primary interacting proteins. BBM was treated as the receptor and the interacting proteins as ligands. The docked score was then reported, and the first 100 conformations were collected and merged into a single file for each docked protein-ligand complex. The Hex tool removed all water molecules and other “hetero” atoms from these input files. Hex then rotates each protein about its coordinate origin and measures the separation between the two origins during the main docking calculation. The docked score was calculated for each orientation, and the highest-scoring orientations were returned as output. PDB files were converted to GRO format (GROMACS format) to perform clustering using GROMACS packages (https://www.gromacs.org/) (accessed on 20 May 2022). The GROMOS cluster method was used with a cutoff of 0.3 nm. The central structure from the most populated cluster was considered representative of the docked complexes.

Post calculation of the docked score, the top 100 conformations were used for further analysis. These 100 conformations were saved and merged into a single file for cluster calculations. The clustering of the trajectories is done to reduce the substantial number of frames in a typical trajectory file to a representative set of distinct frames. The clusters were made from the first 100 conformations in the GROMACS package using the gmx cluster method with an RMS cutoff of 0.3 nm.

### 2.5. Molecular Dynamics Simulation

The Gromacs2021 package (version 2021.5 https://doi.org/10.5281/zenodo.445, Accessed on 20 May 2022 from Texas, USA) was used to perform the molecular dynamic simulation of protein-peptide complexes. Topology parameters were created using CHARMM force fields and allocated to peptides and proteins. The Ewald Particle Mesh method was used to calculate the distant electrostatic force. The neutralization of the system was achieved by adding Na+ and Cl^−^ ions, and the solvation was performed using the TIP3P water cube model. The simulation involved 50,000 steps of minimization, deploying the steepest descent algorithm. The entire system was heated for 500 ps to 310 K under the constant temperature and volume (NVT) ensemble. Post NVT equilibration, the system was processed under the constant temperature and pressure (NPT) ensemble for 1 ns at 1 bar of constant pressure. The SHAKE method was used to limit all hydrogen bonds, a 50 ns (50,000 ps) production simulation run was performed, and the coordinates were stored every 2 ps during the simulation trajectory. During the production run, temperature coupling was performed using the V-rescale method, which is the modified Berendsen thermostat with a time constant of 0.1 ns both for the protein and peptide. Pressure coupling was deployed using the Parrinello-Rahman method. Short-range van der Waal force and electrostatic cut-off were fixed at 1.2 nm. The root means square deviation (RMSD) and root mean square fluctuation (RMSF) were used to examine the conformational variation and structural stability.

### 2.6. Binding Affinity (ΔG) and Dissociation Constant (Kd)

PDB files were uploaded to the PRODIGY server (https://bianca.science.uu.nl/prodigy/) (accessed on 28 May 2022), and the results were collected for binding affinity (ΔG) and dissociation constant (Kd) of the representative structure. PRODIGY counts the number of interfacial contacts (ICs) made at the inner protein complex within a 5.5 Å distance threshold. The server also classified these residues based on polar/nonpolar/charged characteristics.

### 2.7. Interfacial Residues, Peptide Preparations, and Scoring

The representative structure was uploaded to the InterProSurf server (http://curie.utmb.edu/usercomplex.html) (accessed on 1 June 2022) to detect the residues at the interface of the protein-protein surface. The longest continuous stretch of amino acids within the interacting proteins was collected with an allocation for a single missing residue during the selection, which was then replaced with Glycine (G). The peptides ≥ 8 aa in length were selected and modelled into a 3D structure using the APPTEST server (https://research.timmons.eu/apptest) (accessed on 2 June 2022). The primary sequence of the peptide chain was uploaded to the APPTEST server, and the result was stored as PDB files for each structure. PRODIGY server presented a list of residues interacting at the interface of the protein and peptide complex. In BBM, the two continuous stretches of amino acids responsible for DNA-binding (a) residues 210 to 276 and (b) residues 312 to 370 and those that interacted with proteins were used to score the BBM-peptide complexes. Solvent accessible surface area (SASA) was estimated for the protein structure in both the bound (containing peptide) and unbound states using the InterProSurf server. Later, the ∆SASA (Solvent Accessible Surface Area) for each residue was calculated for the central structure of the docked complex. The ΔSASA > 0 implies the presence of residues at the interface.

## 3. Results

### 3.1. BBM Orthologs, Conservation, and Sub-Cellular Localization

A total of 506 proteins among the 6,360,776 sequences in the whole plant proteome database were identical to AtBBM, sorted based on the e-value and percentage identity (Appendix A). The closest orthologs were from the Arabidopsis species (*A. helleri* and *A. lyrate*), followed by many wild species from the Brassicaceae family. The most distant relatives were wild and domesticated soybean (Fabaceae), followed by tree species. These 506 protein sequences were aligned using ClustalW (Appendix A), and a phylogenetic tree was constructed from this alignment.

AtBBM has two important DNA-binding domains, which belong to AP2/ERF family and one disordered region. The AP2/ERF domains have significant conservation across all the 506 orthologous sequences. The midpoint rooted tree showed (Figure 2) that AtBBM is closely related to *Arabidopsis helleri*, *Boechera stricta*, *Capsella rubella*, *Malcolmia maritima*, *Alyssum Lin folium*, *Descurainia Sophia*, and *Corippo islandica*, all of which belong to Brassicaceae. Although *Isatis tinctoria* (dyer’s woad weed) belongs to the Brassicaceae family, it appears to be distantly related to AtBBM along with two ferns, *Ceratopteris richardii* and *Pharus latifolius*, and *Thuja placata*, the western red cedar.

AtBBM is known to localize in the nucleus in *A. thaliana* [22], and the sub-cellular localization predicted by WoLFPSORT annotates these BBM proteins in non-model plants. The results, as summarized in the table below, indicate that majority of these proteins localize in the nucleus, whereas a small number of these proteins could transverse the cytoplasm-nucleus interface, and an even smaller number potentially localize in the mitochondria or chloroplasts (Appendix A).

### 3.2. Protein-Protein and Protein-Ligand Interactions

#### 3.2.1. Ligand Binding Site and Ligand Predictions

The COACH server identified biomolecules such as Chlorophyll-a (CLA), Zn, Fe, c2F, OHX, SO4, streptolydigan, methionine sulfoxide, etc., and unknown proteins that are potential ligands interacting with AtBBM. Further, their active sites were also predicted. The ligands that scored 0.04 or higher on the C-Sore indicate a more reliable prediction, and they are Zn, Fe, CLA, and the unknown protein, all of which are depicted in Figure 3 visualized in iCn3D [39].

#### 3.2.2. Sequence-Based Protein Interactions

Protein interactions, as predicted by STRING, indicate major transcriptional factors such as WUS, LEC1, AGL15, and MYB, among others, to be potential partners of AtBBM, as indicated in Figure 4a. These transcription factors play a key role in embryogenesis and cellular differentiation. Further, BioGRID provided experimentally determined physically interacting proteins BRAHMA (BRM), TPR1, and TPR3, as shown in Figure 4b.

### 3.3. Docking

#### 3.3.1. Docking (BBM-Interacting Proteins) and Docked Conformation Clustering

Using a standalone Hex protein docking tool, the BBM protein was docked with the primary interacting proteins—AGLI5, LEC1, WUS, and LEC2. Both fragments of the BBM protein, fragments 1 and 2, were docked sequentially with the primary interacting proteins of BBM. Here, AGLI5 also shows two fragments. Therefore, these two fragments of AGLI5 were separately docked with each fragment of the BBM protein. The docked scores are listed in Table 2 with their respective receptors and ligands. The average docked score for fragment 1 of the BBM protein was −647.12, and its highest scoring docked complex was with AGLI5 fragment 1, with a docking score of −754.3. Here, the average docked score for fragment 2 of the BBM protein was −612.7, and the AGLI5 fragment 1 complex with fragment 2 of the BBM protein showed the highest docked score of −795.3. The worst among these docked scores was −464.0 for the complex AGLI5 fragment 2 with BBM fragment 1. Similarly, in BBM fragment 2, the worst docked score was −506.2, with the AGLI5 fragment 2. Of the clusters made from the first 100 conformations in the GROMACS package using the gmx cluster method with an RMS cut-off of 0.3 nm, the central structure of the most populated cluster was selected for each of the complexes shown in Table 2. This is the most representative conformation of the cluster centroid.

#### 3.3.2. Binding Energy and Dissociation Constant

The binding affinity indicates the strength of the interaction between the receptor and the ligand, and the dissociation constant indicates the affinity of the receptor for the ligand. The higher the dissociation constant lower is the strength of the bound complex. The binding affinity ΔG and dissociation constant Kd calculated from the PRODIGY server are shown in Table 3 with their respective central structure numbers. The best binding affinity of −12.3 Kcal/mol was reported for the central structure “dock3” of the BBM fragment 1 and LEC1 complex. However, in fragment 2 of BBM, the highest binding affinity was −9.7 Kcal/mol for the central structure “dock61” with AGLI5 fragment 1. Here, the dissociation constants for the best-performing central structures were 8.8 × 10^−10^ M and 7.5 × 10^−8^ M, respectively.

Interatomic contacts (ICs) at a protein-protein complex interface correlate with the experimental binding affinity, largely influencing the interaction [40]. The number of ICs within a 5.5 Å distance threshold is presented in Table 3.

#### 3.3.3. Interfacial Residue Stretch

The results of InterProSurf were used in the construction of a continuous stretch of solvent-exposed residues. These amino acid residues of the interacting proteins with the ∆ SASA score were plotted in the bar graph shown in Figure 5.

#### 3.3.4. Peptide Preparation

Henceforth, the continuous stretch of the amino acid residues of the interacting proteins at the interface of the BBM proteins was analyzed. These stretches of amino acid residues with their respective lengths were listed in Table 4 and Table 5 for BBM fragment 1 and 2, respectively. In addition, the peptides with a stretch of amino acid residues ≥ 8 were marked bold in Table 4 and Table 5 with an X as a single missing residue. These missing residues were replaced with glycine (G) to construct the final peptide sequence. These peptides were: (a) P1: (GFDNYGDPLGVF), (b) P2: (EVAVIVFGKGG), (c) P3: (KNGFYWGQN), (d) P4: (LGRGVGPK), (e) P5: (WGNNGSGM), (f) P6: (RIMGKTGP), and (g) P7: (VAVIVFGK). The peptides P1, P2, P4, P5, and P6 were extracted using BBM fragment 1 protein complex, where P3 and P7 were extracted from the BBM fragment 2 protein complex. P1 and P6 peptides were extracted from LEC1 protein, P2 and P7 were from AGLI5 fragment 1 protein, P4 and P5 were from LEC2, and P3 was from WUS protein. Later, the peptide sequences were submitted to the APPTEST server for 3D structure modeling. The best 3D conformation for each peptide is shown in Figure 6.

#### 3.3.5. Docking and Clustering

The best-docked scores for each protein-peptide pair are listed in Table 6. As explained earlier, the same clustering protocols were applied using the first 100 conformations from the docked structures. The central structures from the most populated cluster are listed in Table 6 for each of the peptide and BBM protein fragment complexes. Peptide P1 showed the best Hex docked score of −470.9, while P7 had the worst docked score. Figure 7 shows the best binding pose of these peptides with their respective BBM fragments.

#### 3.3.6. Protein-Peptide Binding Energy

The central structure of these docked complexes of BBM-peptide complexes was submitted to PRODIGY for binding energy calculation. The binding affinity ΔG and dissociation constant Kd calculated from the PRODIGY server are presented in Table 6 with their respective central structures. Moreover, the PRODIGY server reported the ICs for polar: polar and non-polar:non-polar for the peptides and the BBM fragment protein, as shown in Table 6. The best ΔG −8.1 Kcal/mole was shown by P4, while P3 also scored close to it at −8.0 Kcal/mole. Here, the P7 peptide showed relatively very low binding energy with ΔG −4.9 Kcal/mole. Moreover, all peptides from P1 to P5 had ΔG ≤ −7 Kcal/mole, which was suggested for the strong binding. The dissociation constant followed the same order where P1-P5 had very close Kd values in the order of 10^−6^, which is ten times lower than P6 and P7.

#### 3.3.7. Scoring of Peptides

Eventually, the residues at the interface of the protein-peptide complexes were also detected using the PRODIGY server. These BBM residues were matched with the known DNA-binding site residues (210–276) and (312–370) and belonged to fragments 1 and 2, respectively. The interacting residues of BBM that overlapped with these binding sites were counted and shown in Table 7. The lengths of each of the residue stretches were also mentioned in Table 6. Peptide P1 formed a complex with BBM fragment 1 and exhibited the maximum number of overlapping residues, 13, from the DNA binding site 210–276. Followed by P1, P3, and P7, showed 12 overlapping DNA binding site residues (312–370) at fragment 2 of the BBM protein. P2 and P6 showed 10 overlapping DNA binding residues, each in their respective docked complexes.

#### 3.3.8. Molecular Dynamics Simulation

Molecular docking applied in this study was a rigid method where protein and peptide were not in motion. Rigid docking brings a remarkable limitation to the interaction profile. This can be addressed by applying molecular dynamic (MD) simulation to the docked pose. Here, the P2 (dock10) and P4 protein-peptide complexes are in an explicit MD simulation for 50 ns. P2 had two equally populated clusters (14 members), while other peptides had only one largest cluster. This made P2-protein complexes more structurally uniform than other peptides, enabling their selection for MD simulation. However, P2 (dock10) had a higher number of binding site residues at the interface than P2 (dock89). So, the dock10 pose was preferred for MD simulation, while P4 was also selected based on the binding energy. The protein-peptide system (P2-dock10 and P4) was first brought to 310 K and 1 bar pressure using NVT and NPT ensembles during the equilibrium phase. However, there was fluctuation from 1 bar in the pressure, but it falls under the acceptable range. The RMSD and RMSF analysis provides critical information about the stability and flexibility of the protein-peptide complexes.

#### 3.3.9. Root Mean Square Deviation (RMSD)

Peptide: Root mean square deviation (RMSD) measures the change in structure when two structures are superimposed. Here, the equilibrated structure was considered the reference structure, and all the structures during the 50 ns simulation were compared with the reference structure. After every 20 ps, the simulation trajectory structure coordinates were stored. However, RMSD calculation used the coordinates after every 10 ps. Figure 8a shows the RMSD for the protein-peptide complexes for the peptides P2 (dock10) and P4, respectively. Here, the protein was used as a reference for fitting to calculate the peptide RMSD, and thus the translational motion of the ligand (peptide) was also captured. RMSD re-ported for the peptide is for all atoms taken for the complete trajectory. The peptide structure was most stable in the P2 (dock10) complex, where the RMSD was stabilized at around (0.4–0.6) nm till the end of the 50 ns simulation compared with the equilibrated structure. However, in the P4 complex, the peptide showed the highest RMSD deviation compared with the equilibrated structure. It crossed 2 nm in the beginning and reached 8 nm at 17 ns of the simulation. Then it dropped to 5–6 nm at the end of the simulation. Moreover, the RMSD pattern for the P4 peptide fluctuated during the complete simulation and did not show higher stability in one conformation. Overall, the simulation indicated higher stability of peptide P2 (dock10) in the complex with BBM fragment 1; consistent behavior of peptide P2 (dock10) confirmed the stable binding of peptide P2 (dock10) to the binding cavity of the protein. However, the higher deviation and high fluctuation of the P4 peptide in the complex indicated unstable binding of the peptide P4. This demonstrated the effect of the binding of peptides on the protein molecule.

Figure 8b shows the RMSD for the protein BBM Fragment 1 in the peptide P2 (dock10) and P4 complexes, respectively. Here, all atoms of protein (BBM fragment 1) were used to fit the frames, and RMSD was also calculated for all atoms across the complete trajectory of 50 ns. The BBM Fragment 1 structure was relatively stable in the P4 complex, where mostly the RMSD ≈ 0.4 nm, except for the P2 complex at 25 ns, which crossed 0.4 nm. However, in the P2 (dock10) complex, the BBM Fragment 1 showed a marginally higher RMSD than P4. Moreover, the consistency pattern for the protein in both complexes was very similar and un-distinguishable. Overall, the simulation indicated similar stability of the protein BBM Fragment 1 for both complexes.

#### 3.3.10. Root Mean Square Fluctuation (RMSF)

RMSF values were calculated for peptides in protein-peptide complexes to estimate the individual fluctuation of each residue. Here, the RMSF was calculated for the complete 50 ns of the simulation, and the average fluctuation for each residue for all its atoms was recorded. These are small peptides with 11 and 8 residues. P2 (dock10) had three residues (Residue 1, 9, and 11) with RMSF > 0.3 nm, where the highest RMSF was reported by terminal residue (Residue 11, 0.4 nm). In the P4 complex, there were five residues (Residue 1, 3, 5, 7, and 8) with RMSF > 0.3 nm, but the maximum RMSF was 0.39 nm for the terminal residue (Residue 8). Similar to peptide RMSF calculation, protein RMSF was also calculated for all atoms for each residue averaged over 50 ns simulation. RMSF for the peptide is shown in Figure 9a. Similarly, RMSF for the protein BBM Fragment1 was calculated for both complexes P2 (dock10) and P4. The protein in both complexes showed a similar trend of RMSF, as shown in Figure 9b. In the P4 complex, there were ten residues from protein with RMSF > 0.3 nm, whereas residue stretch 278–285 had continuous residues with RMSF > 0.3 nm. In P2 (dock10) complex, BBM fragment 1 protein had 22 residues with RMSF > 0.3 nm, and there were two continuous stretches 208–212 and 274–285 with consistent RMSF > 0.3 nm.

#### 3.3.11. Binding Energy (Simulation Best Cluster)

Six clusters formed in the P2 (dock10) complex, while in P4, there were 62 clusters. The most populated cluster of P2 (dock10) consisted of 1410 structures. However, the P4 complex had 948 structures in the most populated cluster. The central structure of the two most populated clusters was submitted to PRODIGY for binding energy calculation. The PRODIGY server was used to find the cluster structures’ binding energy ΔG and dissociation constant Kd. The binding affinity ΔG and dissociation constant Kd calculated from the PRODIGY server is shown in Table 8 with their respective cluster structures. The ΔG −6.9 Kcal/mole was shown by P2 (dock10). The P4 peptide showed relatively lower binding energy with ΔG −6.1 Kcal/mole. The dissociation constant showed a 10-fold difference between the peptides binding.

Interaction Residues: The central structures for the best cluster for P2 (dock10) and P4 were used for detecting the binding site residues using the LigPlot server, as shown in Figure 10 and Figure 11 for P2 (dock10) and P4 complexes, respectively.

P2 (dock10) formed three hydrogen bonds with LEU246, GLU279, and GLU285. As the ligand molecule P2 is larger than P4, many hydrophobic contacts (VAL296, TYR295, TYR245, GLY247, SER278, PRO276, LEU261, TYR249, and VAL284) formed in the complex. The interaction map indicates the bigger binding cavity that a peptide can accommodate. Atoms of BBM residues under 4 Å were analyzed for potential hydrophobic contacts. Heavy atoms observed under this range were: TYR295 (main chain: N and side chain: CD, CG, CE, CZ, OH), VAL296 (main chain: CA, N), TYR245 (main chain: C, O and side chain: CD, CE), GLY247 (main chain: CA, N), SER278 (main chain: CA, C, N and side chain: CB, OG), PRO276 (main chain: C, O and side chain: CB), LEU 261 (side chain: CG, CD), TYR249 (side chain: CD, CE, CZ, OH), and VAL284 (side chain: CB, CG).

However, Figure 11 shows the binding site residues of BBM Fragment 1 protein that interact with peptide P4. P4 formed two hydrogen bonds with GLU281 and GLU279. Here, GLU281 formed 2 H-bonds, while GLU279 formed a single H-bond. The GLU281 is a salt bridge interacting with side chains charged at neutral pH. Herein, the ligand molecule P4 is smaller than P2 (dock10), so there are only two hydrophobic interactions (SER278 and GLU283). Hydrophobic contacts were further categorized into the main chain and side chain atoms. These atoms from the hydrophobic contact residues are SER278 (main chain: C, O, and side chain: OG) and GLU283 (main chain: CA, C, O, N, and side chain: CB).

## 4. Discussion

The Baby Boom protein, containing an AP2/ERF domain, is a major transcription factor in plants involved in diverse growth and metabolic pathways [41]. The 584 amino acid length protein, also known as the AP2-like ethylene-responsive transcription factor, has two DNA binding domains (AP2 regions) and one disordered region, which has been reported to be conserved in families such as the Brassicaceae, Poaceae, and Pinaceae [42]. In this study, BBM-like proteins were identified across *Viridiplantae* and *Rhodophyta*, which includes all green algae and plants and red algae. The 506 highly similar proteins ranged from 191 to 746, with an average length of 372 amino acids. AtBBM was reported to be conserved with 98 to 99% sequence identity with certain Brassica species, while the average conservation in these regions was at least 60% across all the orthologous plant sequences [42]. This study’s phylogenetic and domain conservation analyses also indicated the highest similarity with Brassicaceae species. Certain Brassica species with the highest similarity to AtBBM have features that are of significance to the plant breeding community, such as *B. stricta*—apomixes has been widely studied, *B. rapa*—model plant for anther culture and embryogenesis, *Schrenkiella parvula*—extremophyte model that thrives under salt, drought, flooding, chilling, high light, and heat stresses, and *Eutrema salsugineum* (formerly *Thellungiella halophila*) a salt-tolerant relative of both the genetic model Arabidopsis thaliana (Arabidopsis) and agriculturally important members of the genus Brassica. Understanding the structure and interactions of BBM in these plants can enable exploiting these plants to breed superior crop varieties.

BBM is known to localize in the nucleus, although recent reports suggest a translocation between the nucleus and cytoplasm [43]. Since it is a DNA-binding transcription factor, localization in the nucleus is natural, and the same consensus was observed across AtBBM orthologs. Predicting protein-ligand interactions could not only provide insights into BBM protein’s functions but also alter the expression and/or DNA interaction, ultimately leading to the manipulation of BBM for inducing embryogenesis or improving transformation efficiencies. The ligands with the most confidence identified in this study, such as SO4, are sulfate or tetraoxidosulfate (2-), C2F levomefolic acid or metafolin, and Ohx osmium (III) hexamine, and Chlorophyll A is all known to play critical roles in growth and metabolism. For example, Levomefolic acid regulates important cellular functions such as DNA biosynthesis, gene expression regulation, amino acid synthesis and metabolism, and myelin synthesis and repair, whereas other ligands listed are directly or indirectly involved in photosynthesis and seedling development. Light and temperature have a significant role in cell and tissue culture methods. Cold treatment of anthers at 4 °C in the dark induced androgenesis in chickpeas [44]. The predicted interaction of BBM with Chlorophyll A serves as a confirmation of the effect of light on developmental processes in vitro. Red and blue lights have varying effects on the growth and regeneration of plants, and further exploration of these interactions and possible investigation in tissue culture media could have a significant impact on transformation and embryogenesis.

One of the prominent features of BBM is its interaction with major transcriptional factors such as WUS, LEC1, AGL15, and MYB, among others. All these proteins regulate embryogenesis and development via complex gene regulatory networks where various crosstalk and feedback loops play a major role. The immediate next step would be to run a simple pull-down assay to establish the physical interaction of BBM with these other key regulators. From the PPI predictions in this study, BBM was found to interact with BRAHMA (BRM), a large chromatin-remodeling protein with key roles in all stages of plant development [44]. Although there is evidence of physical interaction between BBM and BRM, this is the first report to present that data. Further experimental investigation of this interaction can define the central role played by BBM and BRM in regulatory networks and gene expression.

The determining factor in PPI is the amino acid residues in the binding sites. A number of distinct groups identified the interacting surfaces of a protein with a precision of up to 80% based on the unbound structure [45,46]. Docking simulation, known to identify the correct binding surface by a combined analysis of a whole range of parameters, including solvation potential, amino acid composition, conservation, electrostatics, and hydrophobicity, has been applied in this study. In the first report for BBM PPI, the binding site residues at the interface of PPI and DNA-binding have been identified specifically in a BBM-peptide complex. Such information is fundamental to determining the crystal structure of a BBM and also manipulating protein interactions. Full-length AtBBM can be produced at a lab scale in tobacco cell lines such as the BY2 suspension cultures, where the protein can be isolated, purified, and crystallized to determine the 3-D structure using X-ray crystallography. Creating a mutation in the amino acids could alter their orientation and half-life, impacting binding affinity and proving regulatory mechanisms [47]. In addition, molecular docking results predicted a stretch of amino acids from AGL15 interacting with BBM1. A higher number of hydrophobic amino acids at the interface in the complexes BBM-frag2: LEC1 and BBM-frag2: AGLI5-frag1resonate the concept that non-polar (or hydrophobic) residues predominantly occur at the protein interface, playing a major role in contributing to the driving force for binding [48]. Further, the protein interaction mediating the K-domain of AGAMOUS (AG) is known to contain three hydrophobic alpha helices, which are conserved across its interacting partners like MADS, LEC1, and LEA, key regulators of embryogenesis [49].

These data are valuable for experimental studies and in developing synthetic peptides or peptide mimetics that mimic these interactions, a novel avenue for understanding and manipulating BBM’s mode of action. The addition of these peptide mimetics in the media could have a significant impact on embryogenesis and transformation efficiency. Zhang and co-workers demonstrated that using the basic structure of a protein, information regarding its neighboring structures, co-expression data, and functional and evolutionary similarity generates a scorable predictive power on a genome-wide scale [50]. Along similar lines, the current study is comprehensive in silico analyses of BBM’s sequence and structure with rigorous design deciphering its interactions and ligands. In conclusion, this study presents novel data regarding BBM’s structure, ligand, and protein interactions that should be further explored experimentally to fully exploit BBM’s potential as a transcription factor, thereby helping overcome bottlenecks in plant breeding. Moving forward, in vitro experiments would validate BBM’s physical interactions through pull-down assays, determine the initial binding strength of these peptides by performing Surface plasmon resonance (SPR), and incorporating the peptide mimetics in culture media as elicitors for an embryogenic response. The long-term goal would be to generate sufficient quantities of BBM protein in suspension cultures to purify and crystallize in its native form to solve its crystal structure.

## Figures and Tables

**Figure 1 biomolecules-12-01633-f001:**
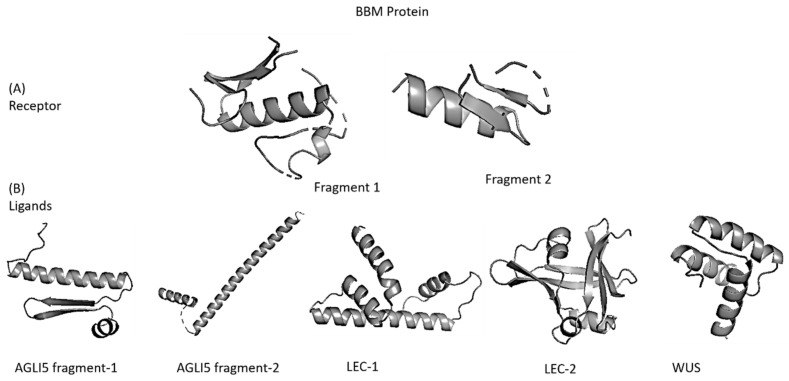
Fragments of BBM and its interacting partners that showed pLDDT > 90 on AlphaFold server used further for docking as (**A**) receptor and (**B**) ligands.

**Figure 2 biomolecules-12-01633-f002:**
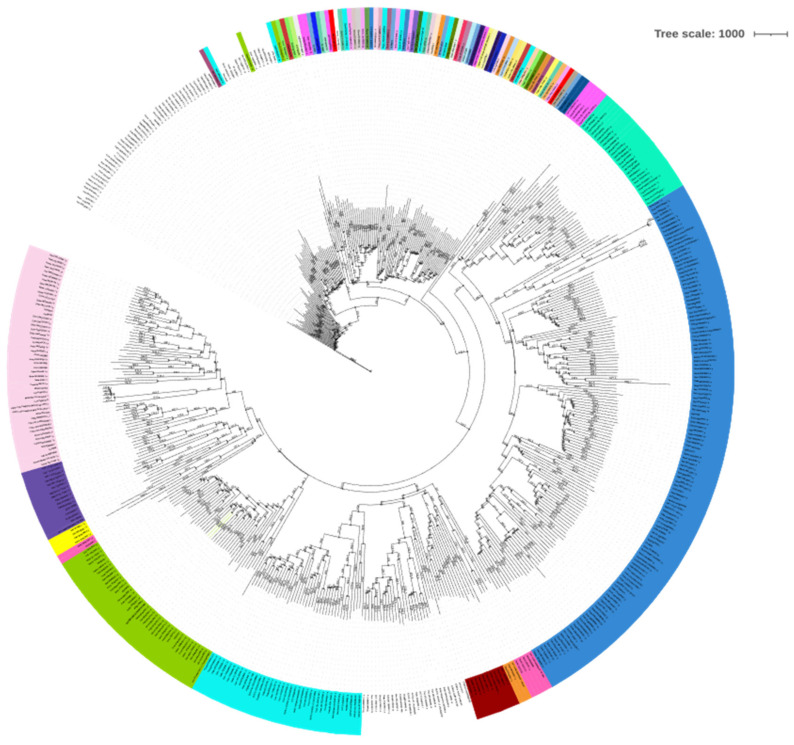
Phylogenetic analysis of BBM. Protein sequences were aligned by Clustal W, and the mid-point rooted phylogenetic tree was constructed using iTOL by the Neighborhood Joining (NJ) method. Based on the phylogenetic relationships, different subgroups were marked with different colors.

**Figure 3 biomolecules-12-01633-f003:**
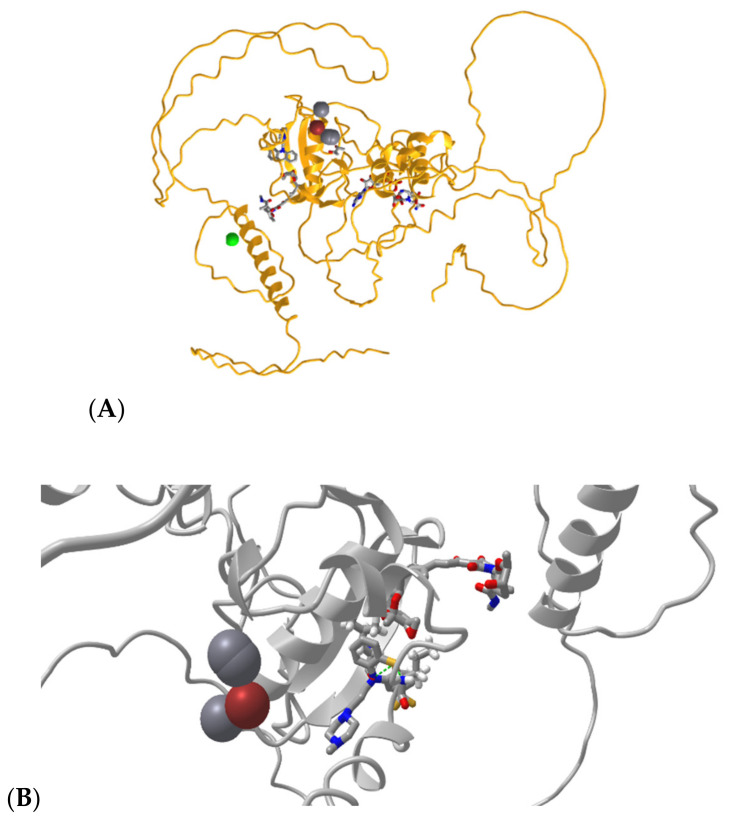
(**A**) Full-length BBM protein with ligand interactions at predicted active sites. (**B**) BBM-ligand interactions with focus on Zn (magenta ball), CLA (grey balls), STD, and MET1.

**Figure 4 biomolecules-12-01633-f004:**
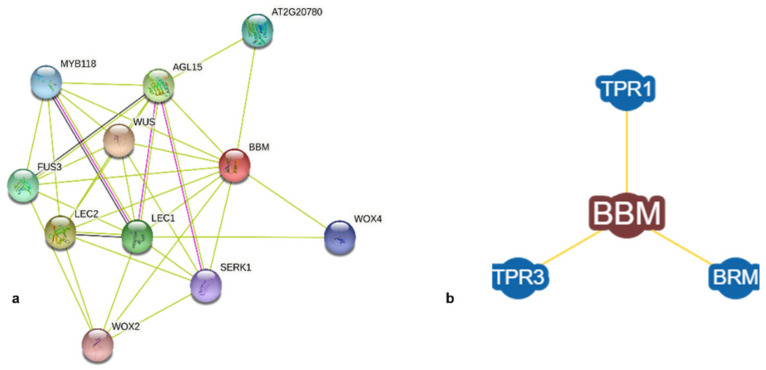
BBM PPI. (**a**) Predicted interactions from STRING. (**b**) Predicted interactions from BioGRID.

**Figure 5 biomolecules-12-01633-f005:**
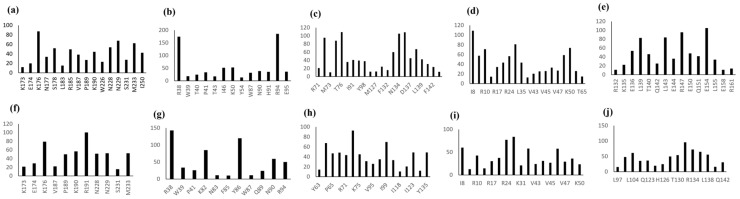
List of the residues for the interaction partners of BBM that showed ∆ SASA (solvent accessible surface area) > 0 in the best-docked complexes (**a**) BBM frag1-LEC2, (**b**) BBM frag1-WUS, (**c**) BBM frag1-LEC1, (**d**) BBM frag1-AGLI5 frag1, (**e**) BBM frag1-AGLI5 frag2, (**f**) BBM frag2-LEC2, (**g**) BBM frag2-WUS, (**h**) BBM frag2-LEC1, (**i**) BBM frag2-AGLI5 frag1, (**j**) BBM frag2-AGLI5 frag2.

**Figure 6 biomolecules-12-01633-f006:**
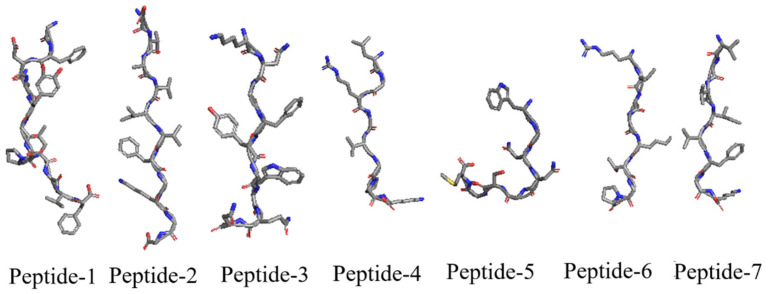
Best 3D conformation of selected 7 peptides (P1–P7) modeled using APPTEST server. Peptides are shown from top to bottom as N-terminal to C-terminal.

**Figure 7 biomolecules-12-01633-f007:**
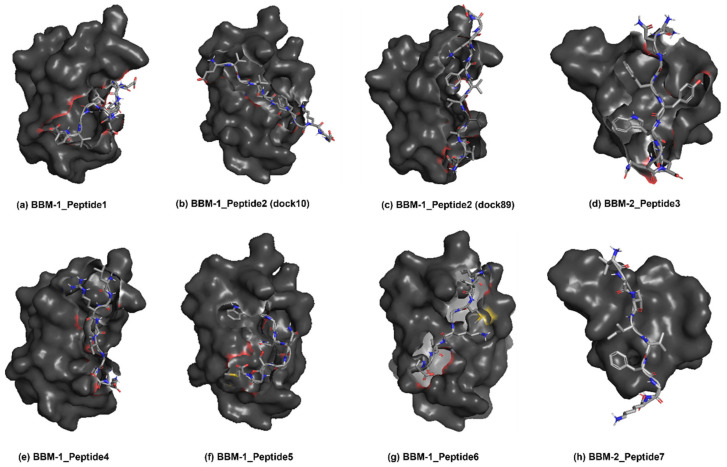
Best docked pose of (**a**) P1-BBM1 (**b**) P2-BBM1 (dock10) (**c**) P2-BBM1 (dock89) (**d**) P3-BBM2 (**e**) P4-BBM1 (**f**) P5-BBM1 (**g**) P6-BBM1 and (**h**) P7-BBM2 complexes.

**Figure 8 biomolecules-12-01633-f008:**
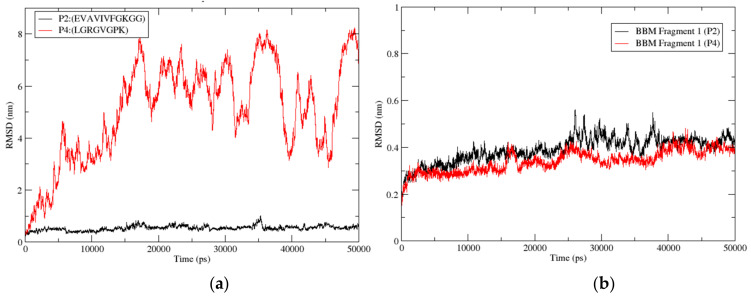
(**a**) Root mean square deviation (RMSD) for peptides P2 (dock10) and P4 was calculated from the protein-peptide complex. (**b**) Root mean square deviation (RMSD) for protein BBM Fragment 1 calculated from the protein-peptide complex. Figures are generated using the xmgrace tool of Linux.

**Figure 9 biomolecules-12-01633-f009:**
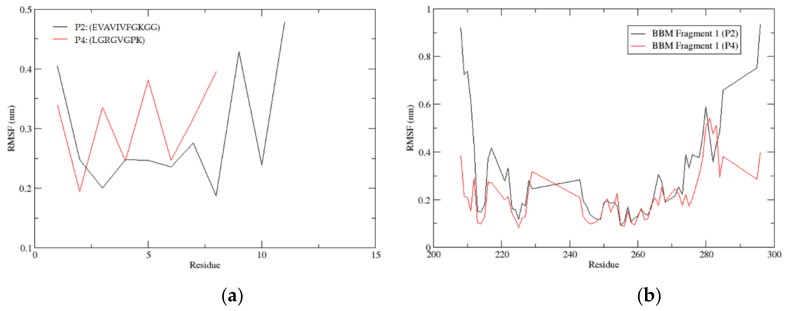
(**a**) Root mean square fluctuation (RMSF) for peptide in protein-peptide complexes with P2 (dock10) and P4. (**b**) Root mean square fluctuation (RMSF) for residue of BBM Fragment 1. Figures are generated using xmgrace tool of linux.

**Figure 10 biomolecules-12-01633-f010:**
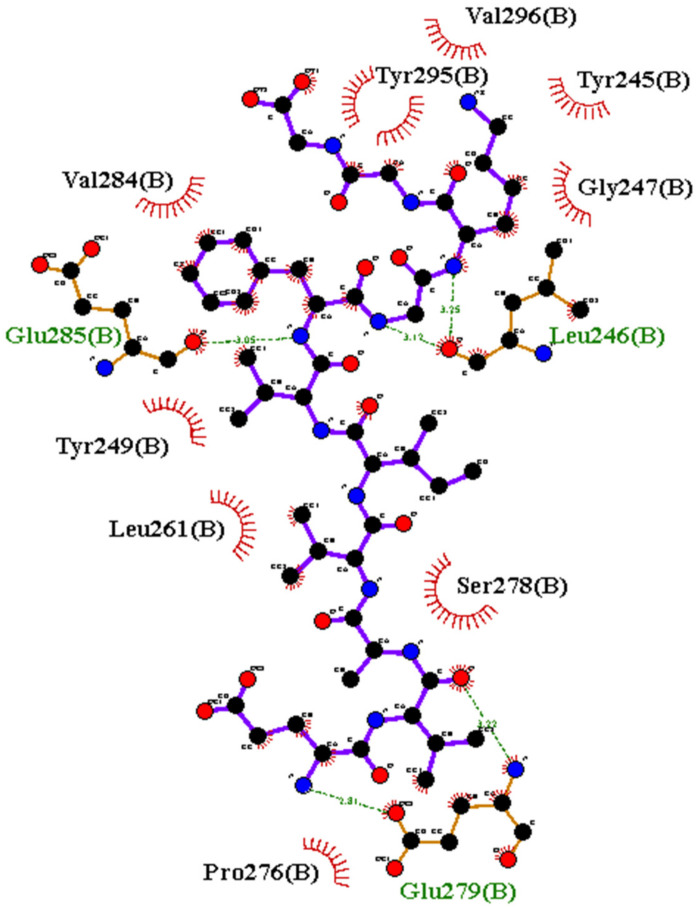
Binding site residues of BBM Fragment 1 protein with the peptide P2 (dock10). Interactions are calculated and figures are generated using LigPlot + 2.2.5.

**Figure 11 biomolecules-12-01633-f011:**
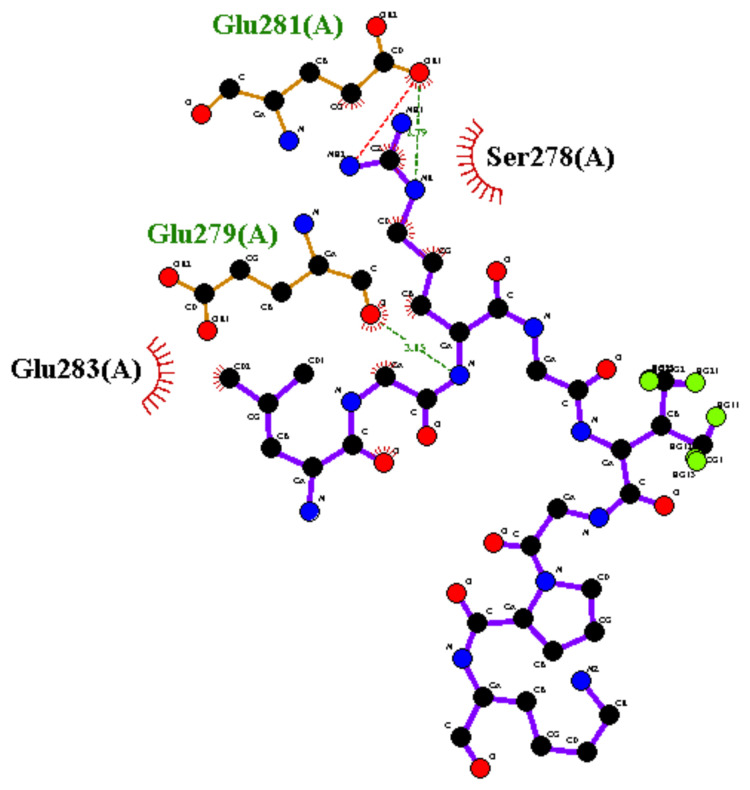
Binding site residues of BBM Fragment 1 protein with the peptide P4.

**Table 1 biomolecules-12-01633-t001:** Primary partners of BBM, their Uniprot IDs, and interaction score from the String database.

Proteins	UniProt ID	Score
AGLI5	Q38847	0.901
LEC1	Q9SFD8	0.900
WUS	Q9SB92	0.908
LEC2	Q1PFR7	0.903

**Table 2 biomolecules-12-01633-t002:** Docked score of BBM fragments with its interacting partners using Hex tool.

Receptor (BBM Protein)	Ligand (Interacting Proteins)	Hex Docked Score
BBM-frag1	LEC2	−670.7
BBM-frag1	WUS	−624.5
BBM-frag1	LEC1	−722.1
BBM-frag1	AGLI5-frag1	−754.3
BBM-frag1	AGLI5-frag2	−464.0
Average score		−647.1
BBM-frag2	LEC2	−554.0
BBM-frag2	WUS	−600.9
BBM-frag2	LEC1	−607.1
BBM-frag2	AGLI5-frag1	−795.3
BBM-frag2	AGLI5-frag2	−506.2
Average score		−612.7

**Table 3 biomolecules-12-01633-t003:** The binding affinity and the dissociation constant values with their respective central structure number and interfacial contact between BBM and interacting proteins.

Receptor (BBM Protein)	Ligand (Interacting Proteins)	Central Structure	Binding Affinity ΔG (kcal mol^−1^)	Dissociation Constant Kd (M)	Polar: Polar	Non-Polar: Non-Polar
BBM-frag1	LEC2	dock46	−11.2	5.7 × 10^−9^	5	12
BBM-frag1	WUS	dock45	−10.6	1.7 × 10^−8^	0	9
BBM-frag1	LEC1	dock3	−12.3	8.8 × 10^−10^	4	11
BBM-frag1	AGLI5-frag1	dock86	−11.1	7.7 × 10^−9^	3	9
BBM-frag1	AGLI5-frag2	dock28	−10.1	4.0 × 10^−8^	0	3
BBM-frag2	LEC2	dock28	−8.6	4.8 × 10^−7^	0	7
BBM-frag2	WUS	dock7	−9.5	1.0 × 10^−7^	0	15
BBM-frag2	LEC1	dock33	−8.7	4.1 × 10^−7^	2	32
BBM-frag2	AGLI5-frag1	dock61	−9.7	7.5 × 10^−8^	0	31
BBM-frag2	AGLI5-frag2	dock2	−8.5	5.9 × 10^−7^	5	14

**Table 4 biomolecules-12-01633-t004:** Continuous stretches of amino acid residues of the interacting partner proteins at the interface of the BBM protein for fragment 1. Selected stretches are shown in bold.

Residue Number	Length
LEC2	
K173, E174, X, K176, N177, S178	6
**L183, X, R-185, X, V-187, X, P-189, K190**	8
**W226, X, N228, N229, X, S231, X, M223**	8
I250	1
WUS	
R38, W39, T40, P41, X, T43	6
I46	1
K50	1
Y54	1
W87	1
N90, H91	2
R94, E95	2
LEC1	
**R71, I72, M73, X, K75, T76, X, P78**	8
I91	1
V95	1
Y98	1
I123	1
M127	1
**G131, F132, D133, N134, Y135, X, D137, P138, L139, X, V141, F142**	12
R145	1
AGLI5-fragment 1	
I8, K9, R10, I11	4
R17	1
F21	1
R24	1
L28	1
L35	1
**E42, V43, A44, V45, I46, V47, F48, X, K50, X, G52**	11
T65	1
AGLI5-fragment 2	
R132	1
K135, E136	2
L139, T140, X, Q142, L143, E144	6
R147	1
E150, Q151	2
E154, L155	2
E158	1
R161	1

The bolded amino acids represent continuous stretches >8 and the X marked in red are the missing residues.

**Table 5 biomolecules-12-01633-t005:** Continuous stretches of amino acid residues of the interacting partner proteins at the interface of the BBM protein for fragment 2. Selected stretches are shown in bold.

Residue Numbers	Length
LEC2	
K173, E174, X, K176	4
V187, X, P189, K190, R191	5
N228, N229, X, S231, X, M233	6
WUS	
R38, W39, X, P41	4
**K82, N83, X, F85, Y86, W87, X, Q89, N90**	9
R94	1
LEC1	
Y63, M64, P65	3
N68	1
R71, I72	2
K75, T76	2
V95	1
Y98, I99	2
T103	1
I118	1
A120	1
I123	1
M127	1
Y135	1
L139	1
F142, I143	2
Y146	1
AGLI5-fragment 1	
I8, K9, R10, I11	4
R17	1
F21	1
R24	1
L28	1
K31	1
L35	1
**V43, A44, V45, I46, V47, F48, X, K50**	8
AGLI5-fragment 2	
L97	1
H101	1
L104, Q105	2
Q123, L124, X, H126, A127	5
T130, V131	2
R134, K135	2
L138, L139	2
Q142	1

The bolded amino acids represent continuous stretches >8 and the X marked in red are the missing residues.

**Table 6 biomolecules-12-01633-t006:** Docked score of the peptide and BBM protein complex with its binding energy, dissociation constant, polar-polar, and nonpolar-nonpolar contacts’ number.

Peptides	Docked Score	Central Cluster	Binding Affinity ΔG (kcal mol^−1^)	Dissociation Constant Kd (M)	Polar: Polar	Nonpolar: Nonpolar
P1	−470.9	dock16	−7.1	6.0 × 10^−6^	0	6
P2	−379.1	dock89	−7.7	2.1 × 10^−6^	0	20
		dock10	−7.2	5.4 × 10^−6^	0	22
P3	−445.5	dock49	−8.0	1.5 × 10^−6^	3	11
P4	−438.4	dock96	−8.1	1.1 × 10^−6^	0	5
P5	−366.4	dock57	−7.7	2.4 × 10^−6^	0	10
P6	−434.9	dock2	−6.7	1.2 × 10^−5^	0	15
P7	−351.1	dock30	−4.9	2.4 × 10^−4^	0	14

**Table 7 biomolecules-12-01633-t007:** Peptides and the corresponding residues of the BBM protein at the interface overlapped with the DNA binding sites.

Peptides	Central Cluster	Overlapping Residues between DNA-Binding Site (210–276) and Docked Interface	Length	Overlapping Residues between DNA-Binding Site (312–370) and Docked Interface	Length
P1	dock16	ILE210, TYR211, LEU246, GLY247, TYR249, LYS251, GLU253, LYS254, ARG257, ALA258, LEU261, ALA262, PHE275	13	-	-
P2	dock89	ARG216, TYR223, ASP250, LYS251, GLU252, GLU253, LYS254, ARG257, PHE275, PRO276	10	-	-
	dock10	ALA225, TYR245, LEU246, GLY247, GLY248, TYR249, GLU253, LYS254, ARG257, ALA258, TYR259, LEU261	12	-	-
P3	dock49	-	-	VAL316, THR317, TRP325, GLN326, ALA327, ARG328, GLN346, GLU347, ALA350, GLU351, TYR353, ASP354	12
P4	dock96	ARG222, GLY248, TYR249, ASP250, LYS251, LYS254, ALA255, ARG257	8	-	-
P5	dock57	TYR249, ASP250, LYS251, GLU253, LYS254, ARG257, ALA258, LEU261	8	-	-
P6	dock2	TYR249, ASP250, LYS251, LYS254, ARG257, ALA258, LEU261, LEU264, LYS265, GLY268	10	-	-
P7	dock30			THR317, TRP325, GLU351, ASP354, VAL316, GLN346, TYR353, ARG328, ALA349, ILE358, ALA350, ALA357	12

**Table 8 biomolecules-12-01633-t008:** Binding energy and Dissociation constant of the first cluster formed after 20 ns MD simulation.

Protein-Protein Complex	ΔG (kcal mol^−1^)	Kd (M) at 25.0 °C
Cluster-P2 (dock10)	−6.9	9.1 × 10^−6^
Cluster-P4	−6.1	3.5 × 10^−5^

## Data Availability

All the raw sequence data was obtained from publicly available sources, which have been presented in the manuscript at appropriate instances.

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
