# Peer review of "BabyBoom: 3-Dimensional Structure-Based Ligand and Protein Interaction Prediction by Molecular Docking"

_biomolecules, 2022, doi:10.3390/biom12111633_

Round 1
Author Response
I sincerely thank the reviewer for the detailed and meticulous review. All the comments and suggestions have been addressed.

Reviewer 2 Report
Comments and suggestions are reported in the attached file.

Author Response
I sincerely thank the reviewer for recognizing the rigor of the manuscript and the potential impact of the in silico study of BBM in determining the experimental studies. The suggestions have been taken and are duly addressed.

Reviewer 3 Report
The author did pure in silico study of the sequence and conservation of the transcription factor Baby Boom (BBM). Using several online web servers, the author predicted potential ligands and proteins interacting with BBM. Furthermore, the residues involved in the potential binding site were determined with molecular docking and MD simulation. None of the results are confirmed by any experiment. Moreover, the author uses fragments from a predicted structure of BBM and other interacting proteins for docking as receptors and ligands. This will lead to mistakes as some surfaces of the fragments might be unavailable for interactions either due to they are buried inside the whole protein or sterically hindered by other parts of the whole protein. Therefore, the docking results are not solid enough to support any conclusion. I could not suggest publishing these uncertain results in any scientific journals.
Author Response
Thank you for taking the time to review the manuscript. This study was conducted as per standard docking and simulation studies in addition to providing comprehensive sequence and structure analyses of BBM. The results present here would have a significant impact on wet lab experiments and plant breeding in the grander scheme.

Round 2
Author Response
Dear Reviewer,
I thank you for your time in reviewing the manuscript and your attention to detail. Your comments have definitely helped improve the quality of the manuscript. Please find the responses below.
The author replied to my comments and modified the manuscript accordingly. However, there are some minor points that have to be resolved before this manuscript can be considered for publication in Biomolecules. They are listed below.
- Q1: Since some data have been changed upon revision I suggest to accurately perform a check of the text. For example: Paragraph 3.3.7, the last line of this paragraph “The P7 complex had the minimum number of DNA binding site residues of 6 at the interface between protein and peptide.” is no longer correct. Please delete it.
Response: Thank you for identifying this error. The sentence has been duly deleted.
Q2: Please correct “explicit MD simulation for 20 ns” with the new timescale (Paragraph 3.3.8). Also, paragraph 3.3.9, 20 ns simulation should be corrected.
Response: MD simulation time has been changed in all instances.
-Q3: I suggest to perform a grammar check of the text, see for example: Paragraph 3.3.4 “P3 and P7 was…” should be “P3 and P7 were…”, Paragraph 3.3.11 “As the ligand molecule P2 is a larger than P4..” please remove “a”. Paragraph 3.3.11, “Atoms of BBM residue” should be “Atoms of BBM residues”.
Response: Thank you for bringing this to my attention. I scanned the entire document to identify and correct grammatical errors.
-Q4: Paragraph 3.3.11, the sentence “Further, the participation of main chain and side chain atoms from the residues that involve in hydrophobic contacts” does not make sense, probably something is missing
Response: I agree, and this sentence has been deleted.
Thank you.
Reviewer 3 Report
The pure docking results are not solid, experimental data are indeed necessary to support the results in this manuscript.
Author Response
Dear Reviewer,
I thank you for your time reviewing the manuscript. Minor modifications have been made throughout the document to emphasize the importance of this in silico study and also our plans to conduct in vitro experimental studies. Some of the experimental goals such as the crystal structure determination is a long-term commitment that needs time and resources that I hope to secure if this manuscript is published. I have the expertise to perform and oversee both tissue culture studies and protein expression/crystallization for a 3d structure solvation. I personally know plant breeders and biotechnologists who would be interested in exploring peptide mimetics in cell culture. I hope you find the manuscript worth publishing as I have extended the MD simulation studies in the first round of edits. All the in silico methods are rigorous and reproducible.
Thank you for your time and consideration.